# Stakeholders' perceptions of recovery from psychosis in Blantyre, Malawi: Definitions, goals, and interventions

Alex Zumazuma[1]*, Abigail M. Morrison[2], Patrick Nyirongo[1], Isaac Mtonga[1], Joshua Chienda[1], Harriet Akello Tikhiwa[3], Hillary Mortensen[4], Bradley N. Gaynes[4,5], Melissa A. Stockton[6], Jack Kramer[2], Bonginkosi Chiliza[7], Brian W. Pence[4], Kazione Kulisewa[1]

**1** Department of Psychiatry and Mental Health, Kamuzu University of Health Sciences, Blantyre, Malawi, **2** Department of Health Behavior, Gillings School of Global Public Health, University of North Carolina at Chapel Hill, Chapel Hill, North Carolina, United States of America, **3** Tidziwe Center, University of North Carolina Project-Malawi, Lilongwe, Malawi, **4** Department of Epidemiology, Gillings School of Global Public Health, University of North Carolina at Chapel Hill, Chapel Hill, North Carolina, United States of America, **5** Division of Global Mental Health, Department of Psychiatry, University of North Carolina at Chapel Hill School of Medicine, Chapel Hill, North Carolina, United States of America, **6** Department of Psychiatry, Perelman School of Medicine, University of Pennsylvania, Philadelphia, Pennsylvania, United States of America, **7** Department of Psychiatry, Nelson R Mandela School of Clinical Medicine, University of KwaZulu-Natal, Durban, South Africa

* alexzumazuma@gmail.com

## Abstract

Goals for recovery from psychosis are perceived differently among those who have been directly impacted by the illness: people with lived experience, caregivers, community members, and those providing treatment. Such differences can limit effective recovery. Understanding the perceptions and goals of each of these groups and the resources needed in different settings is crucial to delivering treatments leading to recovery. Little is known about these perceptions in low- and middle-income countries like Malawi. To understand different stakeholders' definitions of and goals for recovery, and the necessary resources needed to achieve recovery from psychosis in Malawi, we conducted a qualitative study at an outpatient mental health clinic in Malawi involving in-depth interviews and focus group discussions with key informants (n = 48), including people with lived experience (PWLE) and their caregivers, medical practitioners, traditional healers, and community and religious leaders. A thematic approach was used for analysis. All groups shared similar perspectives on recovery, viewing it as the cessation of psychotic symptoms and a return to premorbid function. The cessation of psychotic symptoms included stopping hearing voices and the return of socially appropriate behaviour. A return to premorbid function was described as being able to fully engage in activities of daily living, social interactions, and occupational responsibilities. In contrast to medical practitioners, PWLE additionally emphasized the importance of stopping medications, in part to avoid side effects

**Data availability statement:** We currently do not have ethical approval to share beyond our immediate research team. The Research Ethics Boards who have imposed this restriction are: University of North Carolina at Chapel Hill (contact: irb_questions@unc.edu); the National Health Science Research Committee (contact: directorgeneral@ncst.mw).

**Funding:** This study was funded by the National Institute of Mental Health (R34MH131234). The funders had no role in study design, data collection and analysis, decision to publish, or preparation of the manuscript. Alex Zumazuma, Abigail M. Morrison, Harriet Akello Tikhiwa, Hillary Mortensen, Bradley N. Gaynes, Brian W. Pence and Kazione Kulisewa received a salary from the funding.

**Competing interests:** The authors have declared that no competing interests exist.

like sedation. Common subthemes on the goal of recovery included self-reliance and reintegration into the community. To achieve recovery and meet the goals for recovery, all stakeholders reported a need for medication adherence, community involvement, government involvement, and counselling. However, PWLE also noted that recovery was not as complete if one was still on medications. Successful recovery requires a multidimensional approach involving different institutions and stakeholders. Community-based rehabilitation programs may help facilitate cooperation and integration of care.

## Introduction

Schizophrenia and related psychotic disorders affect 24 million people worldwide (0.32%) and typically have an onset during the second decade of life [1]. While data on the prevalence of schizophrenia and related disorders are available from other settings, most prevalent studies in Africa are nonspecific for the schizophrenia spectrum and other psychotic disorders. The available data for Africa indicates the lifetime prevalence of psychotic disorders to be 1.0-4.4% [2]. In Malawi, population-based surveys for psychosis or schizophrenia are unavailable, but a hospital prevalence study indicated that 30% of hospital admissions were due to schizophrenia [3]. Globally, only 24% of schizophrenia patients fully recover, 40% have poor outcomes, and the rest have suboptimal outcomes [4]. Most people with schizophrenia, therefore, have a chronic disease, which exacts a huge burden on the health care systems, the people living with the illness, their families, and their communities [5–7]. With the onset of the disorder commonly occurring during the second decade of life, people living with the illness may not achieve their life goals (e.g., being employed or getting married) or contribute productively towards their families and communities [1,8].

Evidence-based interventions for schizophrenia and related psychotic disorders require a holistic approach encompassing biological, psychological, social, and spiritual modalities [1,9–12]. The lack of resources to address psychosis in low- and middle-income countries (LMICs) makes it difficult for people with disorders to have comprehensive treatment, especially for diseases that are chronic in nature and require ongoing support. The challenges include inadequate specialised human resources, medication unavailability, and lack of psychosocial interventions and rehabilitation centers [13–15]. One study conducted at an outpatient psychiatric clinic in Malawi found that 35% of people living with a mental illness were asked to buy medication even though 52% of them were unemployed [16].

While LMICs face a lot of challenges to provide healthcare, including a lack of human resources, community-based rehabilitation (CBR) is a promising intervention to provide healthcare that can be delivered by lay workers, thus reducing the burden on the health system [17]. CBR was initiated by the World Health Organization (WHO) to improve the quality of life for people with disabilities and their families by assisting them in meeting their basic needs and promoting their inclusion and participation in society [18]. Before implementing CBR programs, it is helpful to understand

how the population conceptualizes the disease and what it would mean to recover [19]. This allows the CBR program to align with the population's goals for recovery. Such contextualization can be provided by people with lived experience (PWLE) and other key informants who help support PWLE, including medical practitioners, family members, religious leaders, and community leaders. The diverse backgrounds of these key informants may influence the perception of the causes of psychosis and what it means to successfully recover [20–22]. Different informants may also have different goals for the recovery process and a different understanding of the required resources [23–26].

Intervention approaches for people with psychosis have substantially changed over the last century, which may impact how PWLE and other key informants view psychosis and recovery. A more holistic approach has replaced the traditional model of biomedical treatment that mainly focused on symptom reduction [27,28]. These holistic approaches, however, were developed in Western cultures and may not resonate with other cultures [29]. Culture, education, and religion all contribute to differences in understanding across regions [30]. To ensure culturally acceptable interventions for disorders, such treatments must be informed by an understanding of the perceptions, goals, and resources of these distinct groups. In Malawi, several studies have explored the impact of socio-cultural factors on other illnesses, but only a few have addressed schizophrenia and psychosis [19,31,32].

Our goal for this manuscript is to explore the perception of what recovery means, the goals for recovery, and the necessary resources needed, as voiced by PWLE and these key groups involved in the care of people living with psychosis, to ensure acceptable and contextualised CBR programs. This information, generated from the adaptation phase of a pilot study of a new CBR program to support people with psychosis in Malawi, can help us better tailor these CBR programs for successful integration.

## Methods

### Parent study

This manuscript presents findings from a qualitative study conducted during the formative phase of the ENHANCE study. ENHANCE is a pilot study that aims to improve the ongoing, post-acute care of people living with psychosis through nurse-led community-based rehabilitation. The formative research aimed to gather information to guide the adaptation of a patient-friendly, nurse-led, community-based rehabilitation treatment model for individuals in the community living with psychosis in Blantyre, Malawi. Information included how participants thought about psychosis, its causes, treatment, and recovery pathways, and how a nurse-led community-based program might function in this setting. The methods for the qualitative formative research have been previously published [33] and are described below.

### Setting

The study was conducted at the outpatient mental health clinic of Queen Elizabeth Central Hospital (QECH). QECH is one of five public tertiary hospitals in Malawi. QECH provides outpatient psychiatric care; mental health services are not offered as inpatient care apart from liaison psychiatric services. The clinic serves approximately 600 patients per month from within Blantyre and other surrounding districts in the southern part of Malawi.

Most patients attending the clinic have a diagnosis of schizophrenia or a related psychotic disorder. The clinic is staffed by 2 local psychiatrists, medical registrars in training, and mental health nurses.

### Study population and recruitment

The study recruited patients with an established diagnosis under the schizophrenia spectrum and other psychotic disorders. Only patients who had the diagnosis for over 1 year, were 18 years old and above, and resided in Blantyre were eligible. Nurses identified and referred patients whose medical charts indicated schizophrenia or related diagnoses under the spectrum to clinicians for confirmation and eligibility. Twelve PWLE were recruited to participate. PWLE who

consented to take part in the study were asked for permission to have their caregivers interviewed. Eligible caregivers included adults who cared for and lived with a PWLE participant. Twelve caregivers were also enrolled as participants. The lead nurse and the Head of the Department (HOD) of the clinic were asked to identify mental health practitioners who provide care to people with psychosis and might be available to participate in the interviews. Six medical practitioners were recruited from the clinic, including medical doctors specializing in psychiatry and mental health nurses. All medical practitioners participants had been practicing in the QECH psychiatry department for at least a year. In addition, six community leaders, six religious leaders, and six traditional healers were recruited from the community. For community leaders, permission was sought from traditional authorities, who also helped identify other community leaders in their areas. Leaders from the Traditional Healers Association and religious groups were consulted and asked to identify individuals who might be willing to take part in the study. Religious leaders, community leaders, and traditional healers were recruited from the communities served by the hospital (i.e., within Blantyre). Participants were eligible if they were ≥ 18 years old and were residing within Blantyre.

### Data collection tools and process

The research team developed semi-structured interview guides based in a constructivist epistemology and formative research methodologies. Trained interviewers piloted the guides with PWLE and caregivers. The PWLE and caregivers involved in the pilot interviews were not included in the main study. Following the piloting, in-depth interviews (IDIs) were conducted with PWLE, their caregivers, traditional healers, and medical practitioners, and focus group discussions (FGDs) were conducted with community leaders and religious leaders. Data were collected in March and April 2023 at the QECH outpatient clinic.

IDIs with PWLE and caregivers explored their understanding of psychosis and recovery, health-seeking behaviours, and their personal goals and needs for recovery. IDIs with traditional healers and medical practitioners focused on the same themes, with an emphasis on understanding each group's perspective on their role in psychosis treatment and recovery.

Separate FGDs were conducted with community leaders and religious leaders. The FGDs examined participants' understanding of psychosis, their experience of people with psychosis in their communities, understanding the pathways to care and the roles of different groups in treatment, viewpoints on stigma, and their understanding of the course of the illness and interventions needed for recovery, and the role of the family and community in treatment/rehabilitation.

All interviews were conducted in Chichewa by trained interviewers and took approximately one hour. Interviews were audiotaped, then transcribed and translated into English.

### Analysis

The analysis was conducted using Dedoose 9.2.5 [34]. The research team first read the transcripts, and a thematic codebook capturing emerging themes was iteratively drafted [35–37]. For this analysis, we focused on the themes that included understanding psychosis, perspectives on recovery, goals for recovery, and interventions for successful treatment. After coding, all code reports and transcript excerpts related to recovery and interventions for successful treatment were reviewed by AZ and AM. AZ and AM created matrices from the code reports to identify similarities and differences in the themes across all the groups [35]. AZ and AM compared different perspectives of recovery and treatment across the groups and identified the goals of recovery as described by PWLE and caregivers and the challenges they faced in attaining these goals. A description of the goals of recovery was taken from interviews with medical practitioners and traditional healers. Community leaders and religious leaders described the definition of recovery, and the resources required for recovery.

### Ethical consideration

Ethical approval was sought from the University of North Carolina IRB (UNC IRB: 22–1507) and the Malawian National Health Sciences Research Committee (NHSRC: 22/08/2988). Participants provided written informed consent before data collection.

## Results

### Participants characteristics

A total of 48 interviews were conducted. Participants ranged in age from 23 to 82. Of the PWLE, seven were diagnosed with schizophrenia, 4 with schizoaffective, and 1 with unspecified schizophrenia (Table 1).

### Main findings

The findings are presented under two main themes: 1) perspectives on recovery, personal goals, and challenges; and 2) requirements for recovery.

**Perspectives on recovery, personal goals, and challenges.** Perspectives on recovery from psychosis were similarly described by PWLE, caregivers, medical practitioners, traditional healers, community leaders, and religious leaders. In general, participants described recovery from psychosis as both the cessation of psychotic symptoms and a complete return to premorbid functioning, or a more holistic return to "normal." The cessation of psychotic symptoms included the resolution of disorganised behaviour and speech, and behaving in a socially appropriate manner when in public (e.g., wearing clothing in public). A return to premorbid function was described as being able to fully engage in activities of daily living, social interactions, and occupational responsibilities. One caregiver stated:

"*Recovery would be my brother behaving appropriately and engaging in activities that bring benefits to his life, like being able to work and earn an income on his own*" (Caregiver 2, Age 26, Male, Secondary education)

This caregiver described recovery from psychosis as a return to normalcy and attainment of daily functioning, including occupation. He further described recovery as the ability to live an independent life by earning an income independently.

One community leader further described recovery from psychosis by saying:

**Table 1. Participants' characteristics.**

|  | PWLE | Caregivers | Traditional Healers | Medical Practitioners | Community Leaders | Religious Leaders |
|---|---|---|---|---|---|---|
|  | 12 | 12 | 6 | 6 | 6 | 6 |
| **Gender** |  |  |  |  |  |  |
| Female | 3 | 8 | 3 | 4 | 1 | 3 |
| Male | 9 | 4 | 3 | 2 | 5 | 3 |
| **Average Age (Range)** | 37 (23-49) | 45 (24-58) | 61 (42-82) | 37 (27-56) | 62 (54–69) | 52 (48–57) |
| **Education** |  |  |  |  |  |  |
| None | 0 | 1 | 0 | 0 | 0 | 0 |
| Primary | 2 | 4 | 4 | 0 | 4 | 0 |
| Secondary | 8 | 5 | 2 | 0 | 1 | 1 |
| Tertiary | 2 | 2 | 0 | 6 | 1 | 5 |
| **CGI-Score** |  |  |  |  |  |  |
| 1 | 5 |  |  |  |  |  |
| 2 | 4 |  |  |  |  |  |
| 3 | 2 |  |  |  |  |  |
| 4 | 1 |  |  |  |  |  |
| **Diagnosis** |  |  |  |  |  |  |
| Schizophrenia | 7 |  |  |  |  |  |
| Schizoaffective | 4 |  |  |  |  |  |
| Unspecified schizophrenia | 1 |  |  |  |  |  |

*"When we say someone has recovered from psychosis, we mean that the patient has returned to his normal senses, for instance, if he was eating food from the bins and now, he has stopped, if he was speaking abnormal things and now, he is speaking normal things, if he was walking naked and now, he dresses when he walks, yeah, and he can go to work.."* *(Community Leader 4, Age 70,Male, Tertiary education)*

This community leader described recovery as a return to "normal senses" and specified that recovery included both a stop to psychotic symptoms like speaking abnormally and a reintegration into the community in the form of going back to work. A medical practitioner was particularly emphatic that recovery from psychosis included the cessation of hallucinations and delusions, as stated in the quote.

*"When we say that a person has recovered from a mental disorder, it means that the symptoms that they were experiencing, like the hallucinations, stop. The person stops experiencing the symptoms that they were experiencing when their mental disorder was just starting. Even other beliefs that the person with the mental disorder had, beliefs that made them delusional, also stop. They no longer have these stubborn beliefs that they used to stand firm on in the past when their mental disorder was just starting."* *(Medical practitioner 4, Age 28, Male, Tertiary education)*

The medical practitioner understood recovery from psychosis to be an end not just to hallucinations but to delusions as well. All participants emphasised the importance of recovery as a comprehensive process that involved multiple aspects of a patient's life, not just symptom management.

Beyond the cessation of symptoms, one PWLE also described no longer needing medication as a necessary part of recovery from psychosis.

*"To stop taking medication. It is not nice to be taking medication every evening. What if I start working and I oversleep because I took the medication? But I know I need to take medication to go to work and do my job properly."* *(PWLE 11, Age 32, Female, Tertiary education)*

This PWLE believed that recovery from psychosis is not merely the cessation of symptoms and reintegration into the community, but also no longer needing medication to maintain normalcy. Medication is used in the process of recovery to help reduce psychotic symptoms and allow one to reintegrate into society, but a full recovery would no longer involve relying on medication to stop symptoms..

**Goals of recovery.** The goals of recovery focused primarily on helping PWLE reintegrate into society. Caregivers' goals for recovery were in line with their definition of recovery and included both the cessation of symptoms and a return to premorbid daily functioning. One caregiver explained,

*"The long-term goal is that he should be able to complete his studies and be employed. He should be self-reliant, as you know, in life, and currently, I am the one supporting him. But that he should be self-reliant in the future, that people should be able to have confidence in him, that they could employ him and be organized, so that he should be able to take care of himself, and those are my long-term goals."* *(Caregiver 3, Age 52, Female, Secondary education)*

This caregiver highlighted that the goal of recovery was to have his relatives reach independence through the completion of his studies and securing a job. Being self-sufficient was the goal of recovery.

As with the definition of recovery, PWLE also reported that being able to stop taking medications was a personal goal for recovery, which was not reported as a goal by caregivers. One PWLE said:

*"I want to recover completely so that I should never take this medication (for psychosis) again in the future, and also, I would want to recover completely so that I should be independent."* *(PWLE 1, Age 24, Male, Secondary education)*

Both PWLE and the caregiver described the desire to attain independence and normal functionality in occupational, social, and other aspects of life. Further, this PWLE described their desire to stop taking medication as a goal for complete recovery. While specific goals (e.g., returning to employment) were mentioned, it was ultimately the concept of a return to independence and self-reliance that was wanted.

Like PWLE and family members, medical practitioners' goals for recovery included helping PWLE to be able to function in occupational, social, and other areas of their day-to-day life. One medical practitioner explained that,

*"Our goal is that we should see the person being productive in their lives, even though they might still have psychosis. But we are happy, or our goal is that we should see that person being productive in their lives. If they are in school, we should also see that they are also excelling, if they work, we should see that they are being productive at work, that they are dressing decently according to how they might afford to, that they can afford to eat properly and also live like any other person; and in that way we feel good." (Medical practitioner 1, Age 55, Female, Tertiary education).*

The medical practitioner explained that the goal for recovery is to have PWLE be productive, live a happy life, and excel in all their endeavors, even in the presence of symptoms. The medical practitioners' own goals, apart from PWLE's desire to stop medication, aligned with PWLE's goals and emphasized that the attainment of the goals does not require a complete resolution of psychotic symptoms.

Apart from goals for the PWLE, traditional healers indicated personal goals that included a desire to be recognized for their role in recovery. One said,

*"The patient should have a long life so that he can testify that he was treated by such and such traditional healer, yes, in so doing, our reputation is widespread. Yes, so eventually people will be flocking to us seeking treatment, yes, so that is our main goal." (Traditional healer 1, Age 43, Male, Primary education)*

Apart from helping PWLE attain functionality in all domains, traditional healers mentioned getting recognition. The quote by the traditional healer narrated the goal for patients to be treated for their psychosis and wanting them to live a long life. It, however, also stressed a desire to be known for their treatment and to get more patients for treatment. Unlike all other groups whose focus and goals for recovery were centered around the PWLE, traditional healers' focus and goals included their own needs and desires.

**Challenges.** PWLE had clear goals for recovery, but they faced challenges that forced them to change or abandon these goals. The challenges described were mainly the effect of the illness on their functionality, as stated by one PWLE,

*"Everyone has their plans in life, we plan that once I do such and such, I should be somewhere. For example, if I had finished school, I could have been a doctor or a soldier, or I could have been an engineer. So, all these plans changed in my life because of this disease (psychosis) I have. However, when we interact with the health care workers here at the hospital, they advise us not to think too much; instead, they encourage us to be socializing with our friends or reading the bible, going to church, yes, doing things like that." (PWLE 3, Age 36, Female, Secondary education)*

The illness has affected PWLE functionality (i.e., school performance), leading them to abandon or change their goals. Their medical practitioner even encouraged them to revise their recovery goals and focus on shorter-term ones that might be easier to reach.

**Requirements for recovery.** All groups reported multiple needs to achieve recovery and meet the goals for recovery. All groups agreed that medical care and medication adherence were critical to recovery and called for government investment in human resources and medication access. The places where people could seek treatment were not limited to hospitals, and traditional healers emphasized their treatment. One traditional healer was quoted as saying,

*"The successful treatment is getting the treatment either from the healthcare workers, or the traditional healers, and they should receive the proper medication and make sure they do not miss the medication at any single moment. Yes, because the treatment from the traditional healers and the healthcare workers is just the same; for instance, if in their treatment they must, say, receive sixteen injections of drugs, it means they do not have to miss any of that. Likewise, from the traditional healers, the patients need to make sure that they do not miss any medication from the traditional healer. Yes, so in the end, the patient can recover from their illness." (Traditional healer 3, Age 76, Male, Primary education)*

The traditional healer recognizes the critical role of seeking care, receiving appropriate medication, and medication adherence in recovery. Further, the traditional healer suggests that traditional healing is as valid a treatment option alongside conventional medical care. By stating that their methods are like those of health workers, the traditional healer aims to validate traditional healing practices and integrate them within the broader healthcare framework.

Community leaders and PWLE described believing that treatment from traditional healers relied on honesty and an individual's belief in the efficacy of the treatment. One community leader cast doubt on the effectiveness of a traditional healer's treatment, especially since they believed most traditional healers are "scammers."

*"Successful treatment from a traditional healer requires that the traditional healer be an expert in his job so that the treatment he provides should help the patient recover. The traditional healer should also be honest; if his treatment cannot help the patient heal, he should tell the caregivers of the patient with psychosis that he has failed, and they should seek care elsewhere. But I would say it's not guaranteed that you will get successful treatment from the traditional healers because most of the traditional healers are scammers and they cannot provide a successful treatment, while others are honest and eventually help the patient recover, but not all traditional healers help." (Community Leader 4, Age 70, Male, Tertiary education)*

One PWLE believed and thought that the effectiveness of traditional healers' treatment depends on personal beliefs. He illustrated how his father was cured using traditional medicine, even though he, himself, did not believe in traditional medicine.

*"I am not aware of any other ways, although others say that by using traditional medicine, a person can have their mental disorder cured. However, I really do not believe this. I had my father, and he was also suffering from the same mental disorder as me. He used traditional medicine, and he was cured. So, it all depends on your beliefs. My beliefs and the way that I see it are that it is better to follow the guidance that one receives from the health service providers." (PWLE 2, Age 23, Male, Secondary education)*

The PWLE indicated that personal beliefs have an impact on the effectiveness of treatment from traditional healers, i.e., those who believe they will be cured of their illness. Together, the views of the traditional healers, community leaders, and PWLE suggest that an inclusive approach to treatment, acknowledging the value of both traditional and conventional medical practices, may be appropriate in the Malawian context.

Beyond medical care and medication adherence, one medical practitioner described the need for a successful treatment as multidimensional, involving health care service providers, family members, and the community. The medical practitioner further suggested the involvement of the workplace, as all these factors play a huge role in a person's recovery.

*"Successful treatment of psychosis involves the health care service providers, the family members of the client, as well as the community members. If that person is employed, it also involves their workplace." (Medical practitioner 1, Age 55, Female, Tertiary education)*

The quote from the medical practitioner indicates that for successful treatment to take place, the community, medical practitioners, and family all need to actively be involved and support the person with lived experiences.

There is a broad consensus among key informants on the essential components of effective treatment for psychosis, which include medical care and medication adherence as well as support from multiple facets of a person's life. This highlights the need for a multidimensional approach to treatment, involving both medical and psychosocial support, and the need for coordinated efforts among healthcare providers, government agencies (e.g., to improve access to medications), and community organizations.

## Discussion

All key informant groups held similar views regarding the definition of recovery, the goals of recovery, and the resources necessary to recover from psychosis in Malawi. All groups defined recovery as a cessation of psychotic symptoms and a return to premorbid functioning. PWLE further discussed being able to stop taking medication as a sign of a complete recovery. The goals of recovery were focused on helping PWLE regain independence and reintegrate into society. The participants also identified the importance of medical care, adherence to medication, and the involvement of the community and government as the necessary resources for PWLE to achieve recovery.

Recovery from psychosis is a multifaceted concept influenced by cultural, medical, and personal perspectives [20–22,30]. Recovery can be related to clinical, functional, and personal outcomes [38]. Clinical recovery has focused on sustained remission of disease symptoms and restoration of functioning, whereas personal recovery resulting from patients has focused on being hopeful and living a satisfying and fulfilling life even in the presence of symptoms [39–42]. These views were reflected in our interviews, as all key informant groups described recovery from psychosis as a resolution of psychotic symptoms and a complete return to premorbid functioning.

When thinking about the clinical definition of recovery, emphasizing sustained remission and a return to premorbid functioning, a recent systematic review and meta-analysis has demonstrated that only 24% of people with psychosis reach full recovery [4]. It may also be useful to think of recovery in terms of personal outcomes. Personal recovery does not rely on symptom remission but rather on living an independent and fulfilling life, which was a clear goal described by both PWLE and their caregiver. A study conducted in India found that patients attributed their recovery to the ability to work and get married [40]. Further, PWLE also described recovery as being off medications. This inclusion resonates with other studies that have demonstrated the desire of patients to be off medication as part of their recovery [43].

Across all groups, recovery goals were described as patient-centered, emphasizing the reintegration of PWLE into society. This approach aligns with the WHO guidance and technical package for community mental health services: "promoting person-centered and rights-based approaches" [44]. Patient-centeredness fosters positive patient outcomes, enhances healthcare quality, ensures compassionate and effective care, and has greater benefits than traditional approaches [45–47]. Of note, the medical practitioners' goals for the patient aligned with the personal definition of recovery (i.e., wanting the patient to be productive and to excel in all undertakings, even in the presence of symptoms). This demonstrated that even though their perception of recovery aligned with the clinical definition, their holistic approach toward the patient aligns with the definition of personal recovery.

Challenges reported by PWLE included symptoms of the illness and side effects of the medication. These findings are like those of a study in India where patients reported that the symptoms hindered the achievement of personal goals, which they included in their definition of recovery (e.g., getting back to work, getting married) [40]. This challenge underlines the importance of psychoeducation. Educating PWLEs and their caregivers about the prognosis, treatment options, and realistic goals can reduce frustration and set achievable expectations. Furthermore, psychoeducation has also been shown to reduce the rate of hospitalization, relapses, and readmissions, and it encourages drug adherence [48,49]. Moreover, adopting a culturally sensitive approach in psychoeducation can bridge the gap between medical, traditional, and religious perspectives. To ensure better outcomes and satisfaction, the psychoeducation programs need to be patient-centered [50,51]

Our analysis describes that a successful recovery includes medical care, access and adherence to medication, and support from the government, family, and community. In the treatment of mental disorders, biopsychosocial approaches are used, and our results aligned with the strategy [1,9–12]. Antipsychotic treatment improves long-term outcomes [52]. However, medication adherence is affected by many factors, i.e., drug factors, disease factors, personal characteristics, problem behaviors, income, and quality of life, with higher levels of family and social support and positive treatment attitudes improving adherence [53]. With many people having low income and low quality of life in Malawi, enhancing protective factors like family and social support and patient-centered psychoeducation can improve overall recovery. Again, studies in Malawi have demonstrated that the duration of untreated psychosis is long (i.e., disease factor) [50]. Early detection through community collaboration can reduce the duration of untreated psychosis, thereby promoting recovery. Government involvement is also paramount to ensure that medication and psychosocial services (e.g., counselling) are available. The places where people could access care were not limited to health facilities but also included traditional healers. This has been established in other studies, which demonstrated that 60% of patients present to traditional healers, a pattern influenced by socio-cultural understanding of the causation of the illness (e.g., witchcraft, spirit possession, and curses) [54,55]. It is crucial to incorporate and collaborate with traditional healers in the delivery of mental health services [56]

Our findings support established conceptual models detailing processes for personal recovery among people with severe mental illness, such as CHIME [57]. CHIME identifies five key components of individual recovery: connectedness, hope and optimism about the future, identity, meaning in life, and empowerment [57–60]. The domain of connectedness is reflected in participants' emphasis on the need for strong support systems among medical professionals, families and community networks as part of recovery. PWLE's and caregivers' aspirations for education, independent living, and future achievements also resonate with the hope component. PWLE indirectly touched on identity and meaning, through a focus on rebuilding their lives, improving their quality of life and having meaningful goals and social roles beyond their illness. PWLE and other cadres also emphasized the importance of regaining control over their lives by taking care of themselves, maintaining their independence, graduating from school, starting businesses, and establishing relationships, which align with the CHIME framework's empowerment component. While our data collection and analysis were not explicitly informed by CHIME, future research unpacking recovery could be strengthened by applying established recovery frameworks.

## Conclusion

Our qualitative research analysis is the first to explore the definition of recovery, the goals of recovery, and the necessary resources required for a successful recovery from psychosis in groups with diverse backgrounds. Our results showed that recovery from psychosis is perceived similarly among diverse groups of key informants and includes both medical care and adherence to medication as well as a return to pre-morbid functioning and daily life. Our results also showed that successful recovery requires support from medical providers as well as family and communities. These findings support the promotion of community-based rehabilitation programs to facilitate recovery.

## Author contributions

**Conceptualization:** Alex Zumazuma, Abigail M. Morrison, Kazione Kulisewa.

**Formal analysis:** Alex Zumazuma, Abigail M. Morrison.

**Funding acquisition:** Kazione Kulisewa.

**Investigation:** Alex Zumazuma.

**Methodology:** Alex Zumazuma, Abigail M. Morrison, Melissa A. Stockton, Kazione Kulisewa.

**Project administration:** Harriet Akello Tikhiwa.

**Writing – original draft:** Alex Zumazuma, Abigail M. Morrison.

**Writing – review & editing:** Patrick Nyirongo, Isaac Mtonga, Joshua Chienda, Hillary M. Mortensen, Bradley N. Gaynes, Jack Kramer, Bonginkosi Chiliza, Brian W. Pence, Kazione Kulisewa.

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
