## [Decision Letter · Decision Letter 0]

11 Nov 2025

PMEN-D-25-00388

DEFINITIONS, GOALS, AND INTERVENTIONS FOR RECOVERY IN MALAWI

PLOS Mental Health

Dear Dr. Zumazuma,

Thank you for submitting your manuscript to PLOS Mental Health. After careful consideration, we feel that it has merit but does not fully meet PLOS Mental Health’s publication criteria as it currently stands. Therefore, we invite you to submit a revised version of the manuscript that addresses the points raised during the review process.

We look forward to receiving your revised manuscript.

Kind regards,

Lambert Zixin Li, Ph.D.

Academic Editor

PLOS Mental Health

Journal Requirements:

1.  Please send a completed 'Competing Interests' statement, including any COIs declared by your co-authors. If you have no competing interests to declare, please state "The authors have declared that no competing interests exist". Otherwise please declare all competing interests beginning with the statement "I have read the journal's policy and the authors of this manuscript have the following competing interests:"

1. Please clarify all sources of funding (financial or material support) for your study. List the grants (with grant number) or organizations (with url) that supported your study, including funding received from your institution.

2. State the initials, alongside each funding source, of each author to receive each grant.

3. State what role the funders took in the study. If the funders had no role in your study, please state: “The funders had no role in study design, data collection and analysis, decision to publish, or preparation of the manuscript.”

4. If any authors received a salary from any of your funders, please state which authors and which funders.

Additional Editor Comments (if provided):

Dear Authors,

Thank you for submitting your manuscript to PLOS Mental Health. The reviewers have provided constructive feedback and identified areas that require further clarification and improvement. Based on their comments and my own assessment, I invite you to revise and resubmit your manuscript for further consideration.

Please address all reviewer comments carefully in both the revised manuscript and a detailed response letter. Ensure that your submission meets the journal’s formatting and reporting standards, and that the language is clear and professional throughout.

We look forward to receiving your revised manuscript.

Sincerely,

Lambert Zixin Li, PhD

Reviewers' comments:

Reviewer's Responses to Questions

**Comments to the Author**

1. Does this manuscript meet PLOS Mental Health’s publication criteria?

Reviewer #1: Yes

Reviewer #2: Yes

2. Has the statistical analysis been performed appropriately and rigorously?

Reviewer #1: N/A

Reviewer #2: N/A

3. Have the authors made all data underlying the findings in their manuscript fully available (please refer to the Data Availability Statement at the start of the manuscript PDF file)?

Reviewer #1: Yes

Reviewer #2: No

4. Is the manuscript presented in an intelligible fashion and written in standard English?

Reviewer #1: Yes

Reviewer #2: Yes

Reviewer #1: This qualitative study highlights how different stakeholders in Malawi conceptualize recovery from psychosis and also highlights an evidence-based CBD approach in a cultural context. The methodology, analysis, and interpretation are sound, and the conclusion is consistent with the data. However, minor grammatical and typographical edits are suggested, like spacing issues. Also, the study needs to clarify the theoretical framework of recovery within established models (CHIME framework) and global recovery scenarios. In the result section, the statements/ quotations of participants need to be simplified.

Reviewer #2: Title: Definitions, Goals, and Interventions for Recovery in Malawi. Suggested revision to ‘stakeholders’ perceptions about recovery from psychosis in Blantyre, Malawi.’

Abstract: Well written.

Introduction: Clear description of the problem.

Materials and methods: The authors talked about the parent study but failed to provide information concerning which aspect was being presented in this particular manuscript. Please make this clear. The authors did not state the study design explicitly nor the technique for recruitment. Please make this clear. Although this was described as a hospital-based study, participants were drawn from the hospital and the community. Authors need to describe the setting better in this regard. The authors also need to give information on specific inclusion and exclusion criteria for their participants. For example, did patients who were recruited need to have a diagnosis for a specific length of time, or were even recently diagnosed patients included? For the caregivers, did they need to live with patients or be involved in their care for a specific period to be eligible or were these not considered? What about the time in the designated role for the practitioners? Authors need to state whether these were considered, and if not, they should consider including the information as limitations. Line 133: has both PWLE and caregivers and PWLE and guardians. Please choose one and stick with it. Authors should include the ethical clearance numbers for both IRBs.

Result and discussion: The authors should label the quotes to give some information regarding who made the quote. Caregiver 2 does not tell me the gender, age, or level of education of the person who made the quote. These may be important when trying to discuss the results. This is the same for all the participants. Line 218 mentions MP. I presume that this means medical practitioner, but the authors need to spell out each acronym before use. The phrase ‘this view may be at odds with views from MP’ is not explained as part of the result. Please consider revising or removing.

**Do you want your identity to be public for this peer review?** For information about this choice, including consent withdrawal, please see our Privacy Policy

Reviewer #1: **Yes:** SIDRA BATOOL

Reviewer #2: No

---

## [Decision Letter · Decision Letter 1]

5 Jan 2026

STAKEHOLDERS’ PERCEPTIONS OF RECOVERY FROM PSYCHOSIS IN BLANTYRE, MALAWI: DEFINITIONS, GOALS, AND INTERVENTIONS

PMEN-D-25-00388R1

Dear Dr. Zumazuma,

We are pleased to inform you that your manuscript 'STAKEHOLDERS’ PERCEPTIONS OF RECOVERY FROM PSYCHOSIS IN BLANTYRE, MALAWI: DEFINITIONS, GOALS, AND INTERVENTIONS' has been provisionally accepted for publication in PLOS Mental Health.

Best regards,

Lambert Zixin Li, Ph.D.

Academic Editor

PLOS Mental Health

Dear Authors,

Thank you for your revision. I am pleased to accept the manuscript for publication.

Sincerely,

Lambert Zixin Li, PhD

Reviewer Comments (if any, and for reference):

Reviewer's Responses to Questions

**Comments to the Author**

Reviewer #1: All comments have been addressed

publication criteria?

Reviewer #1: Yes

3. Has the statistical analysis been performed appropriately and rigorously?

Reviewer #1: Yes

4. Have the authors made all data underlying the findings in their manuscript fully available (please refer to the Data Availability Statement at the start of the manuscript PDF file)?

Reviewer #1: Yes

5. Is the manuscript presented in an intelligible fashion and written in standard English?

Reviewer #1: Yes

Reviewer #1: (No Response)

**Do you want your identity to be public for this peer review?** For information about this choice, including consent withdrawal, please see our Privacy Policy

Reviewer #1: **Yes:** SIDRA BATOOL
